# GLEN: General-Purpose Event Detection for Thousands of Types

**Qiusi Zhan[1]\*, Sha Li[1]\*, Kathryn Conger[2], Martha Palmer[2], Heng Ji[1], Jiawei Han[1]**
[1]University of Illinois Urbana-Champaign
[2]University of Colorado Boulder
{qiusiz2, shal2, hengji, hanj}@illinois.edu
{kathryn.conger, martha.palmer}@colorado.edu

## Abstract

The progress of event extraction research has been hindered by the absence of wide-coverage, large-scale datasets. To make event extraction systems more accessible, we build a general-purpose event detection dataset GLEN which covers 205K event mentions with 3,465 different types, making it more than 20x larger in ontology than today's largest event dataset. GLEN is created by utilizing the DWD Overlay, which provides a mapping between Wikidata Qnodes and PropBank rolesets. This enables us to use the abundant existing annotation for PropBank as distant supervision. In addition, we also propose a new multi-stage event detection model CEDAR specifically designed to handle the large ontology size in GLEN. We show that our model exhibits superior performance compared to a range of baselines including InstructGPT. Finally, we perform error analysis and show that label noise is still the largest challenge for improving performance for this new dataset.[1]

## 1 Introduction

As one of the core IE tasks, event extraction involves event detection (identifying event trigger mentions and classifying them into event types), and argument extraction (extracting the participating arguments). Event extraction serves as the basis for the analysis of complex procedures and news stories which involve multiple entities and events scattered across a period of time, and can also be used to assist question-answering and dialog systems.

The development and application of event extraction techniques have long been stymied by the limited availability of datasets. Despite the fact that it is 18 years old, ACE 2005 [2] is still the de

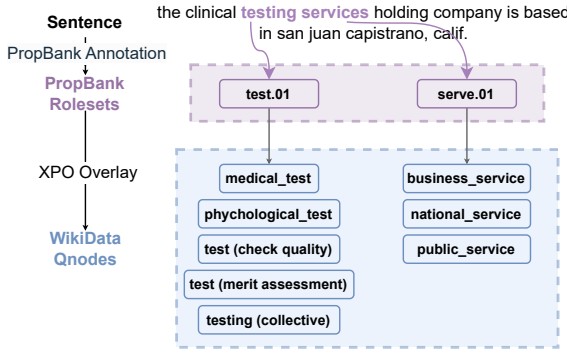

Figure 1: The GLEN event detection dataset is constructed by using the DWD Overlay, which provides a mapping between PropBank rolesets and WikiData Qnodes. This allows us to create a large distantly-supervised training dataset with partial labels.

facto standard evaluation benchmark. Key limitations of ACE include its small event ontology of 33 types, small dataset size of around 600 documents and restricted domain (with a significant portion concentrated on military conflicts). The largest effort towards event extraction annotation is MAVEN (Wang et al., 2020), which expands the ontology to 168 types, but still exhibits limited domain diversity (32.5% of the documents are about military conflicts).

The focus of current benchmarks on *restricted ontologies in limited domains* is harmful to both developers and users of the system: it distracts researchers from building scalable general-purpose models for the sake of achieving higher scores on said benchmarks and it discourages users as they are left with the burden of defining the ontology and collecting data with little certainty as to how well the models will adapt to their domain. We believe that event extraction can, and should be made accessible to more users.

To develop a general-purpose event extraction system, we first seek to efficiently build a high-quality open-domain event detection dataset. The difficulty of annotation has been a long-standing

---

[1]Our dataset, code, and models are released at https://github.com/ZQS1943/GLEN.git.
[2]https://www.ldc.upenn.edu/collaborations/past-projects/ace

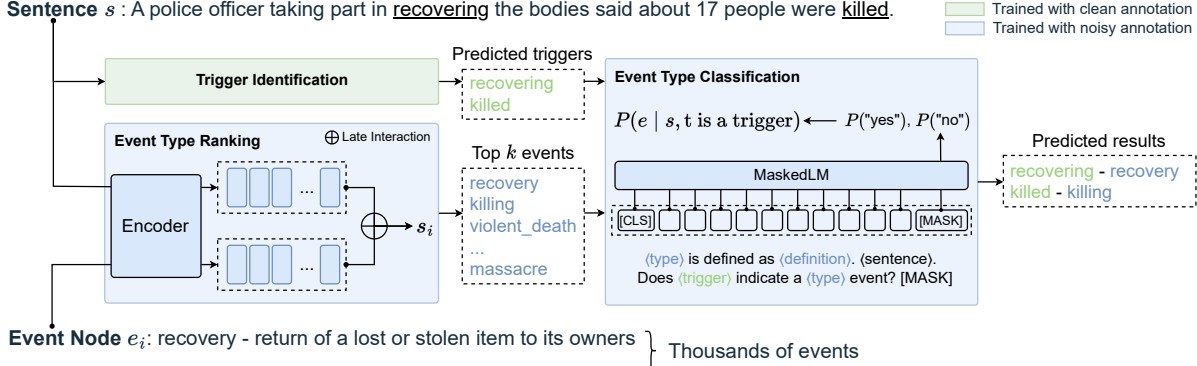

Figure 2: Overview of our framework. We first identify the potential triggers from the sentence (**trigger identification**) and then find the best matching type for the triggers through a coarse-grained sentence-level **type ranking** followed by a fine-grained trigger-specific **type classification**.

issue for event extraction as the task is defined with lengthy annotation guidelines, which require training expert annotators. Thus, instead of aiming for perfect annotation from scratch, we start with the curated DWD Overlay (Spaulding et al., 2023) which defines a mapping between Wikidata[3] and PropBank rolesets (Palmer et al., 2005)[4] as displayed in Figure 1. While Wikidata Qnodes can serve as our general-purpose event ontology, the PropBank roleset information allows us to build a large-scale distantly supervised dataset by linking our event types to existing expert-annotated resources. In this process, we reuse the span-level annotations from experts, while dramatically expanding the size of the target ontology. Our resulting dataset, GLEN(The GeneraL-purpose EveNt Benchmark), covers 3,465 event types over 208k sentences, which is a 20x increase in the number of event types and 4x increase in dataset size compared to the previous largest event extraction dataset MAVEN. We also show that our dataset has better type diversity and the label distribution is more natural (Figures 3 and 4).

We design a multi-stage cascaded event detection model CEDAR to address the challenges of large ontology size and distant-supervised data as shown in Figure 2. In the first stage, we perform **trigger identification** to find the possible trigger spans from each sentence. We can reuse the span-level annotations, circumventing the noise brought by our distant supervision. In the second stage, we perform **type ranking** between sentences and all event types. This model is based on Col-

BERT (Khattab and Zaharia, 2020), which is a very efficient ranking model based on separate encoding of the event type definition and the sentence. This stage allows us to reduce the number of type candidates for each sentence from thousands to dozens, retaining ∼90% recall@10. In the last stage, for each trigger detected, we perform **type classification** to connect it with one of the top-ranked event types from the first stage. We use a joint encoder over both the sentence and the event type definition for higher accuracy.

In summary, our paper makes contributions in (1) introducing a new event detection benchmark GLEN which covers 3,465 event types over 208k sentences that can serve as the basis for developing general-purpose event detection tools; (2) designing a multi-stage event detection model CEDAR for our large ontology and annotation via distant supervision which shows large improvement over a range of single-stage models and few-shot Instruct-GPT.

## 2 Related Work

As the major contribution of our work is introducing a new large-scale event detection dataset, we review the existing datasets available for event extraction and the various ways they were created.

**Event Extraction Datasets** ACE05 set the event extraction paradigm which consists of event detection and argument extraction. It is also the most widely used event extraction dataset to date. The more recent MAVEN (Wang et al., 2020) dataset has a larger ontology selected from a subset of FrameNet (Baker et al., 1998). FewEvent (Deng et al., 2020) is a compilation of ACE, KBP, and

---

[3]wikidata.org

[4]A roleset is a set of roles that correspond to the distinct usage of a predicate.

| Dataset | | #Documents | #Tokens | #Sentences | #Event Types | #Event Mentions |
|---------|---|-----------|---------|-----------|-------------|-----------------|
| ACE2005 | | 599 | 303k | 15,789 | 33 | 5,349 |
| MAVEN | | 4480 | 1276k | 49,873 | 168 | 118,732 |
| GLEN | Train | 5607 | 3641k | 187,468 | 3,464 | 184,806 |
| | Dev | 311 | 204k | 10,359 | 1720 | 9878 |
| | Test | 306 | 201k | 10,627 | 1334 | 8290 |
| | **Total** | **6224** | **4045k** | **208,454** | **3,465** | **205,045** |

Table 1: Statistics of our GLEN dataset in comparison to the widely used ACE05 dataset and the previously largest event detection dataset MAVEN.

Wikipedia data designed for few-shot event detection. A number of datasets (DCFEE (Yang et al., 2018), ChFinAnn (Zheng et al., 2019), RAMS(Ebner et al., 2020), WikiEvents (Li et al., 2021), DocEE (Tong et al., 2022)) have also been proposed for document-level event extraction, with a focus on argument extraction, especially when arguments are scattered across multiple sentences. Within this spectrum, GLEN falls into the category of event detection dataset with a heavy focus on wider coverage of event types.

**Weak Supervision for Event Extraction** Due to the small size of existing event extraction datasets and the difficulty of annotation, prior work has attempted to leverage distant supervision from knowledge bases such as Freebase(Chen et al., 2017; Zeng et al., 2017) and WordNet (Araki and Mitamura, 2018; Tong et al., 2020). The former is limited by the number of compound value types (only ∼20 event types are used in both works) and the latter does not perform any typing. Weak supervision has also been used to augment training data for an existing ontology with the help of a masked language model (Yang et al., 2019) or adversarial training (Wang et al., 2019). Our dataset is constructed with the help of the DWD Overlay mapping which is defined between event types (Qnodes) and PropBank rolesets instead of on the level of concrete event instances (as in prior work that utilizes knowledge bases).

## 3 The GLEN Benchmark

To build the GLEN benchmark, we first build a general-purpose ontology based on the curated DWD Overlay and then create distantly-supervised training data based on the refined ontology (Section 3.1). In order to evaluate our model, we also create a labeled development set and test set through crowdsourcing (Section 3.2).

### 3.1 Event Ontology and Data

The DWD Overlay is an effort to align WikiData Qnodes to PropBank rolesets, their argument structures, and LDC tagsets. This mapping ensures that our ontology is a superset of the ontology used in ACE and ERE (Song et al., 2015). (See Section 3.3 for a detailed comparison of the ontology coverage.)

To make this ontology more suitable for the event extraction task, we remove the Qnodes related to cognitive events that do not involve any physical state change such as belief and doubt. [5] We also discovered that many rolesets such as ill.01 were heavily reused across the ontology (mainly due to the inclusion of very fine-grained types), therefore we manually cleaned up the event types that were associated with these rolesets. We show some examples of removed Qnodes in Appendix Table 8.

Since the DWD Overlay is aligned with PropBank, we propose to reuse the existing PropBank annotations[6]. After the automatic mapping, each event mention in the dataset would be associated with one or more Qnodes, which leads to the **partial label** challenge when using this distantly-supervised data. We then perform another round of data filtering based on the frequency of rolesets (details in Appendix C). After these cleaning efforts, we used the annotation for 1,804 PropBank rolesets, which are mapped to a total of 3,465 event types.

To make our data split more realistic and to preserve the document-level context, we split the dataset into train, development, and test sets based on documents using a ratio of 90/5/5. Note that although our test set is only 5% of the full data, it is already similar in scale to the entire ACE05 dataset. For datasets such as OntoNotes and AMR that con-

---

[5]This is in line with the scope of events as defined in previous ACE and ERE datasets.

[6]https://github.com/propbank/propbank-release

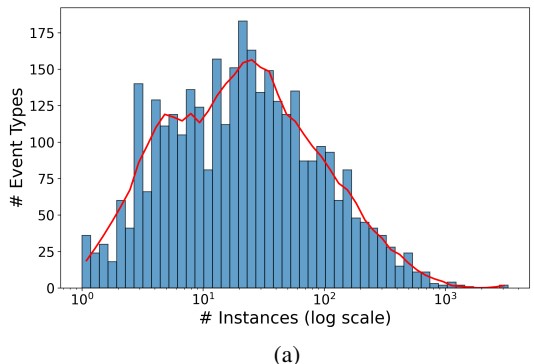 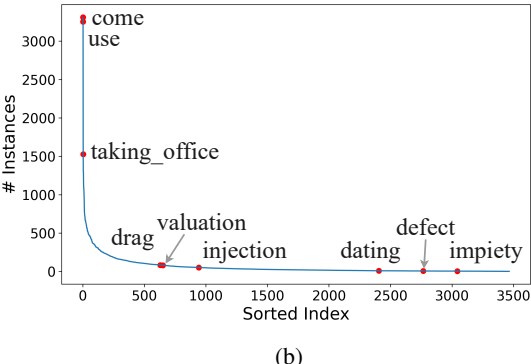

(a)             (b)

Figure 3: The event type distribution of GLEN. In the training set, instances associated with $N$ labels are weighted as $\frac{1}{N}$. Figure (a) illustrates the distribution of event types based on the number of instances. In figure (b), we show the top three most popular event types as well as randomly sampled types from the middle and tail of the distribution curve.

tain documents from multiple genres (newswire, broadcast, web blogs etc.), we perform stratified sampling to preserve the ratio between genres. The statistics of our dataset are listed in Table 1. Compared with ACE05 and MAVEN, GLEN utilizes a 20x larger ontology and 4x larger corpus.

## 3.2 Data Annotation

Instead of performing annotation from scratch, we formulate the annotation task as a multiple-choice question: the annotators are presented with the trigger word in context and asked to choose from the Qnodes that are mapped to the roleset.

For the test set, we hired graduate students with linguistic knowledge to perform annotation. For the development set, we screened Mechanical Turk workers that had high agreement with our in-house annotators and asked those who passed the screening to participate in the annotation. The weighted average kappa value for exact match on the test set (27 annotators), is 0.60, while for the dev set (981 annotators) is 0.37 over 5.2 options. If we allow for a soft match for event types of different granularity, such as `trade` and `international_trade`, the kappa value is 0.90 for the test set and 0.69 for the dev set. For more details on the annotation interface, see Appendix C.

## 3.3 Data Analysis

We first examine our event ontology by visualizing it as a hierarchy (as shown in Figure 4) with the parent-child relations taken from Wikidata (the `overlay_parent` field in DWD Overlay). Our ontology offers a wider range of diverse events, including those related to military, disaster, sports,

social phenomena, chemicals, and other topics, indicating its novelty and potential usefulness.

We show the event type distribution in our dataset in Figure 3. Our type distribution closely mirrors real-world event distributions. The distribution exhibits frequent events such as `come` and `use`, along with a long tail of rare events such as `defect` and `impiety`. In terms of type diversity, for ACE, the single most popular event type `attack` accounts for 28.8% of the instances and the top-10 event types account for 79.4% of the data. Our type distribution is much less skewed with the top-10 events composing 8.3% of the data.

Figure 5 illustrates the part-of-speech distribution of trigger words in our dataset[7]. Over 96% of trigger words are verbs or nouns, which is similar to that of 94% in MAVEN and 90% in ACE2005. In addition, 0.6% of the triggers are multi-word phrases.

## 4 Method

In the event detection task, the goal is to find the trigger word offset and the corresponding event type for every event. The main challenge for our dataset is the **large ontology size** and **partial labels** from distant supervision. To mitigate label noise, we first separate the trigger identification step from the event typing step, since trigger identification can be learned with clean data. Then, to handle the large ontology, we break the event typing task into two stages of type ranking and type classification to progressively narrow down the search space. Finally, in the type classification model, we adopt

---

[7] We use universal POS tagging tools from https://www.nltk.org/.

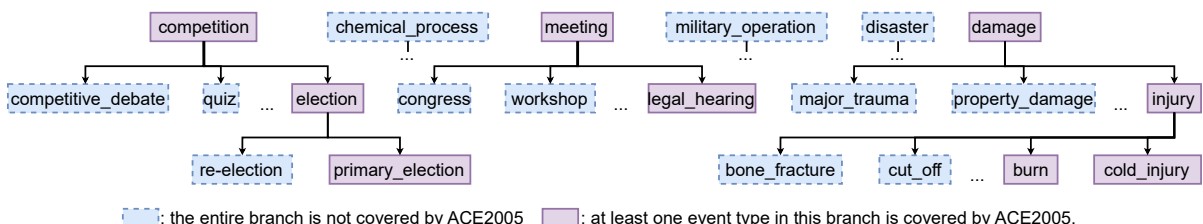

: the entire branch is not covered by ACE2005.    : at least one event type in this branch is covered by ACE2005.

Figure 4: A comparison between our ontology and that of ACE. Our dataset offers broader coverage of events compared to ACE05, with diverse branches ranging from `sports_competition` to `military_operation`.

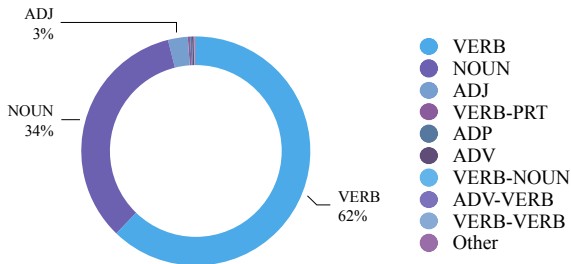

Figure 5: The Part-of-Speech distribution of trigger words for events in GLEN. Multiple POS tags mean that the trigger has multiple words.

a self-labeling procedure to mitigate the effect of partial labels.

## 4.1 Trigger Identification

In the trigger identification stage, the goal is to identify the location of event trigger words in the sentence. This step only involves the sentence and not the event types. We formulate the problem as span-level classification: to compute the probability of each span in sentence $s$ being an event trigger, we first obtain sentence token representations based on a pre-trained language model:

$$[\mathbf{s}_1 \cdots \mathbf{s}_n]^T = \text{PLM}([\text{CLS}]s_1 \cdots s_n[\text{SEP}]) \in \mathbb{R}^{n \times h},$$

where $\mathbf{s}_i$ is a $h$-dimensional representation vector of token $s_i$.

Then we compute the scores for each token being the start, end, and part of an event trigger individually as:

$$f_\square(s_i) = \mathbf{w}_\square^T \mathbf{s}_i, \quad \square \in \{\text{start}, \text{end}, \text{part}\},$$

where $\mathbf{w}_\square \in \mathbb{R}^h$ is a learnable vector. We then compute the probability of span $[s_i \cdots s_j]$ to be an event trigger as the sum of its parts:

$$p([s_i \cdots s_j])$$
$$= \sigma(f_{\text{start}}(s_i) + f_{\text{end}}(s_j) + \sum_{k=i}^{j} f_{\text{part}}(s_k)).$$

The model is trained using a binary cross entropy loss on all of the candidate spans in the sentence.

## 4.2 Event Type Ranking

In the next stage, we perform event type ranking over the entire event ontology for each sentence. Since our ontology is quite large, to improve efficiency we make two design decisions: (1) ranking is done for the whole sentence and not for every single trigger; (2) the sentence and the event type definitions are encoded separately.

We use the same model architecture as Col-BERT (Khattab and Zaharia, 2020), which is an efficient ranking model that matches the performance of joint encoders while being an order of magnitude faster.

We first encode the sentence as:

$$\vec{\mathbf{s}} = \text{PLM}([\text{CLS}][\text{SENT}]s_1 \cdots s_n[\text{SEP}]) \in \mathbb{R}^{(n+1) \times h}$$
$$[\mathbf{h}_s^1, \cdots, \mathbf{h}_s^m] = \text{Norm}(1\text{dConv}(\vec{\mathbf{s}})) \in \mathbb{R}^{m \times h}$$
$$(1)$$

[SENT] is a special token to indicate the input type. The one-dimensional convolution operator serves as a pooling operation and the normalization operation ensures that each vector in the bag of embeddings $\mathbf{h}_s$ has an L2 norm of 1.

The event type definition is encoded similarly, only using a different special token [EVENT].

Then the similarity score between a sentence and an event type is computed by the sum of the maximum similarity between the sentence embeddings and event embeddings:

$$\rho_{(e,s)} = \sum_{h_s} \max_{h_e}(\mathbf{h}_e^T \mathbf{h}_s) \qquad (2)$$

Our event type ranking model is trained using the distant supervision data using a margin loss since each instance has multiple candidate labels. This margin loss ensures that the best candidate is scored higher than all negative samples.

$$\mathcal{L} = \frac{1}{N} \sum_s \sum_{e^-} \max\{0, (\tau - \max_{e \in C_y} \rho_{(e,s)} + \rho_{(e^-,s)})\} \quad (3)$$

$e^-$ denotes negative samples, $C_y$ is the set of candidate labels and $\tau$ is a hyperparameter representing the margin size.

## 4.3 Event Type Classification

Given the top-ranked event types from the previous stage and the detected event triggers, our final step is to classify each event trigger into one of the event types. Similar to (Lyu et al., 2021), we formulate this task as a Yes/No QA task to take advantage of pre-trained model knowledge. The input to the model is formatted as "⟨type⟩ is defined as ⟨definition⟩. ⟨sentence⟩. Does ⟨trigger⟩ indicate a ⟨type⟩ event? [MASK]". We directly use a pretrained masked language model and take the probability of the model predicting "yes" or "no" for the [MASK] token, denoted as $P_{\text{MLM}}(w)$, where $w \in \{\text{"yes"}, \text{"no"}\}$. From these probabilities, we calculate the probability of the event type $e$ as follows:

$$P(e) = \frac{\exp(P_{\text{MLM}}(\text{"yes"}))}{\exp(P_{\text{MLM}}(\text{"yes"})) + \exp(P_{\text{MLM}}(\text{"no"}))} \quad (4)$$

To train the model, we employ binary cross-entropy loss across pairs of event triggers and event types.

As mentioned in Section 3.1, our data contains label noise due to the many-to-one mapping from Qnodes to PropBank rolesets. We adopt an incremental self-labeling procedure to handle the partial labels. We start by training a base classifier on clean data labeled with PropBank rolesets that map to only one candidate event type. Despite being trained on only a subset of event types, the base model exhibits good generalization to rolesets with multiple candidate event types. We then use the base classifier to predict pseudo-labels for the noisy portion of the training data, selecting data with high confidence to train another classifier in conjunction with the clean data.

## 5 Experiments

### 5.1 Experiment Setting

**Evaluation Metrics** Previous work mainly uses **trigger identification F1** and **trigger classification F1** to evaluate event detection performance. An event trigger is correctly identified if the span matches one of the ground truth trigger spans and it is correctly classified if the event type is also correct. In addition to these two metrics, due to the size of our ontology, we also report **Hit@K** which measures if the ground truth event type is within the top-$K$ ranked event types for the event trigger (the event trigger span needs to be correct). This can be seen as a more lenient version of trigger classification F1.

**Baselines** We compare with three categories of models: (1) classification models including **DM-BERT** (Wang et al., 2019), **token-level classification** and **span-level classification**; (2) a definition-based model **ZED** (Zhang et al., 2022) and (3) **InstructGPT**(Ouyang et al., 2022) with few-shot prompting. We attempted to compare with CRF models such as OneIE model (Lin et al., 2020) but were unable to do so due to the memory cost of training CRF with the large label space. For detailed baseline descriptions and hyperparameters, see Appendix B.

### 5.2 Results

We show the evaluation results in Table 2 and some example predictions in Table 3. The first group of baselines (DMBERT/TokCls/SpanCls) are all classification-based, which means that they treat event types as indexes and do not utilize any semantic representation of the labels. We observe that DMBERT will have a substantially lower trigger identification score if we allow spans of more than 1 token to be predicted due to its max pooling mechanism (it will produce overlapping predictions such as "served", "served in", "served in congress"). TokCls's performance hints on the limit for learning only with partial labels. As shown in the example, TokCls usually predicts event types that are within the candidate set for the roleset, but since it has no extra information to tell the candidates apart, the prediction is often wrong.

Although ZED utilizes the event type definitions, it only achieves a minor improvement in performance compared to TokCls. ZED employs mean pooling to compress the definition embedding into a single vector, which is more restricted compared to the joint encoding used by our event type classification module.

We observe that InstructGPT with in-context learning does not perform well on our task. The low trigger identification scores might be attributed to the lack of fine-tuning and under-utilization of the training set. The low classification scores are mainly caused by the restriction in input length [8] which makes it impossible to let the model have full

---

[8]The maximum input length was 2049 tokens at the time of our experiments.

| Model | Trigger Identification | | | Trigger Classification | | | Hit@k | | |
|---|---|---|---|---|---|---|---|---|---|
| | Prec | Recall | F1 | Prec | Recall | F1 | Hit@1 | Hit@2 | Hit@5 |
| DMBERT (Wang et al., 2019) | 56.87 | 84.32 | 67.93 | 32.93 | 48.84 | 39.34 | - | - | - |
| Token Classification | 68.04 | 82.19 | 74.46 | 41.48 | 50.10 | 45.38 | - | - | - |
| Span Classification | 62.36 | 78.71 | 69.58 | 37.36 | 47.16 | 41.69 | - | - | - |
| TokCls + ZED (Zhang et al., 2022) | - | - | - | 40.01 | 56.25 | 46.76 | 56.84 | 60.35 | 60.80 |
| InstructGPT (32 shot) | 28.41 | 42.30 | 33.99 | 11.76 | 17.52 | 14.08 | - | - | - |
| CEDAR w/o self-labeling | - | - | - | 49.21 | 61.62 | 54.72 | 68.21 | 80.21 | 89.18 |
| CEDAR | 71.05 | 88.96 | **79.00** | 50.91 | 63.74 | **56.60** | 71.30 | 80.84 | 88.63 |

Table 2: Quantitative evaluation results (%) for event detection on GLEN.

| Context | Predictions |
|---|---|
| You do **pay** income tax on your paycheck and the sales tax on consumable goods? | CEDAR: payment(transfer of an item of value)
TokCls: disbursement (payment from a public fund)
ZED: income (consumption and savings opportunity)
GPT3: None |
| ... a situation ( brought on mostly by the police ) where information is **discovered** in a way that perpetuates the story | CEDAR: discovery (detecting something new)
TokCls: medical_finding
ZED: physical_finding (from a physical examination of patient)
GPT3: discovery (detecting something new) |
| 40% of the female students at Georgetown law **reported** to us that they struggle financially as a result of this policy. | CEDAR reporting (producing a oral or written report)
TokCls: scoop (journalism term for a new interesting story)
ZED: reporting (producing a oral or written report)
GPT3: None |

Table 3: Comparison across different systems. Correct predictions are shown in green. Predictions that map to the same PropBank roleset are shown in orange. For ZED, we show the TokCls+ZED variant which has better performance.

knowledge of the ontology. As a result, only 57.8% of the event type names generated by InstructGPT could be matched in the ontology. With a larger input window and possibly fine-tuning, we believe that large LMs could achieve better performance.

For our own model, we show that decoupling trigger identification from classification improves TI performance, and performing joint encoding of the definition and the context improves TC performance. Furthermore, using self-labeling can help improve top-1 classification performance by converting partial labels into clean labels.

## 6 Analysis

In this section, we investigate the following questions: (1) Which component is the main bottleneck for performance? (Section 6.1) and (2) Does our model suffer from label imbalance between types? (Section 6.2)

### 6.1 Per-Stage Performance

Our model CEDAR comprises three components: trigger identification, event type ranking, and event

| Component | Hit@k | | |
|---|---|---|---|
| Type Ranking | Hit@10 | Hit@20 | Hit@50 |
| | 89.86 | 93.52 | 95.44 |
| Type Classification (ground truth in top-10) | Hit@1 | Hit@2 | Hit@5 |
| | 78.70 | 89.37 | 97.76 |

Table 4: Evaluation for type ranking and classification separately. The scores for type classification component are computed over the subset of data where the ground truth event is among the top-10 ranked results.

type classification. Table 2 shows precision, recall, and F1 scores for trigger identification, while Table 4 presents the performance of event type ranking and classification. We assess per-stage performance using Hit@k metrics for event type ranking on ground truth trigger spans and event type classification on event mentions where the ground truth event type appears in the ranked results by the type ranker. The scores indicate that the primary bottleneck exists in the precision of trigger identification and the Hit@1 score of type classification.

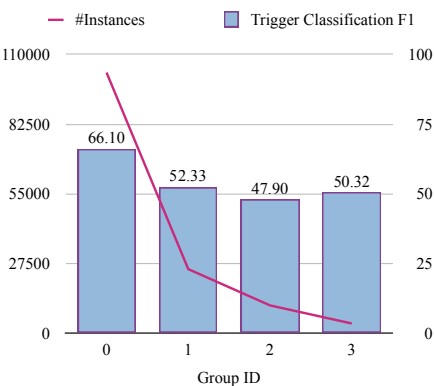

Figure 6: Relationship between the number of instances of different event types in the dataset and our model's performance. The event types are divided into four groups based on their frequencies.

## 6.2 Type Imbalance

Figure 3 illustrates the long-tailed label distribution in our dataset. To investigate whether our model is affected by this imbalance, we divided the event types into four groups separated at the quartiles based on their frequency and calculated the performance per group. The resulting figure is shown in Figure 6. While we do see that the most popular group has the highest F1 score, the remaining groups have comparable scores. In fact, we see two factors at play in defining the dataset difficulty: the ambiguity of the event type and the frequency of the event type. Event types that are more popular are also often associated with rolesets that have a high level of ambiguity, which balances out the gains from frequency.

## 6.3 Error Categories

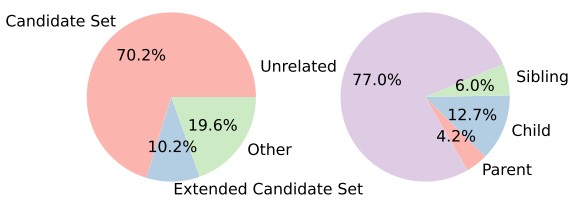

Figure 7: Categorization of errors based on the relation between the predicted event type and the candidate set produced by the mapping (left) and the predicted event type and the ground truth event type on the event ontology hierarchy (right).

We categorize the type classification errors from the CEDAR model based on the relationship between our predicted event type and the ground truth event type as shown in Figure 7 and Table 5.

Most of our errors come from the noisy annotation (**Candidate Set**): our model can predict an event type that falls within the set of candidate types associated with the ground truth PropBank roleset but fails to find the correct one. **Extended Roleset** refers to the predicted event being associated with a roleset that shares the same predicate as the ground truth. The uncategorized errors are often due to the imperfect recall of our event ranking module (as in the second example in Table 5 where the context is long and "fund" fails to be included in the top-10 ranked event types), or cases where our model prediction is related semantically to the ground truth but the event types have no connections in the hierarchy (as in the first example in Table 5 where we predicted "quantification" instead of "measurement"). On the other hand, in another 22.9% of the cases, we predict an event that is close to the ground truth on the XPO hierarchy, with the ground truth either being the child (**Child**), parent (**Parent**), or sibling node (**Sibling**) of our predicted type. This suggests that better modeling of the hierarchical relations within the ontology might be useful for performance improvement.

## 7 Conclusions and Future Work

We introduce a new general-purpose event detection dataset GLEN that covers over 3k event types for all domains. GLEN can be seen as our first attempt towards making event extraction systems more accessible to the general audience, removing the burden on users of designing their own ontology and collecting annotation. We also introduce a multi-stage model designed to handle the large ontology size and partial labels in GLEN and show that our tailored model achieves much better performance than a range of baselines.

Our model CEDAR can be used as an off-the-shelf event detection tool for various downstream tasks, and we also encourage members of the community to improve upon our results, (e.g. tackle the noise brought by partial labels) or extend upon our benchmark (e.g. include other languages).

## 8 Limitations

We perceive the following limitations for our work:

- Lack of argument annotation. Event argument extraction is a critical component of event extraction. At the time of publication, GLEN is only an event detection dataset without argument annotation. This is an issue that we are

| Error Category | Context | Predicted | Gold |
|---|---|---|---|
| Candidate | 3 workers set off a critical **reaction** in 990000 when they poured too much uranium into a precipitation tank. | reaction (response to stimulus) | chemical_reaction |
| | France is urged to increase **research** funding and support for innovations as a way to deal with the problem. | research_method | research (systematic study) |
| Extended Candidate | Regulators and research firms promised that the $1.5 billion **settlement** would be finalized two months ago. | settlement(distortion of a building) | settlement(operations relating to the payment) |
| Child | People **working** for minimum wage are producing a large number of products or services. | work (activities performed as a means of support) | work (activity done by a person for economic gain) |
| Sibling | In the middle east **conflict**, do you think the United States should take Israel's side, take the Palestinians' side, or not take either side? | social_conflict (struggle for agency or power in society) | armed_conflict (conflict including violence) |
| Parent | Doctors will **examine** him for signs that the cancer may have come back while he awaiting trial in a Russian jail. | inspection | physical_examination (process by medical professional) |
| Other | Ironically, in the 90's, "character matters" became a **well-worn** slogan on the right. | wears(clothing or accessory) | wear (damaging, gradual removal or deformation) |
| | In theory, one could argue that the computer models are accurate and that the real **measurements** have some problems. | quantification | measurement |
| | Though **funds** have already been allocated and voted on for the project, Blair himself insists that things are still "very much open"... | voting | fund |

Table 5: Examples of erroneous type predictions. The trigger word (phrase) is shown in **bold**. In some cases, the error falls into multiple categories. We prioritize the XPO hierarchy-related categories since they are rarer.

actively working towards for future releases of the dataset.

- Timeliness of documents. The source documents of the PropBank annotated datasets are not very new. In particular, the OntoNotes dataset contains news articles from 2000-2006 [9]. Hence, there is the possibility of a data distribution shift between our training set and any document that our model is being applied, which might cause performance degrade, as shown in other tasks (Rijhwani and Preotiuc-Pietro, 2020).

- Completeness of ontology. Although our dataset is the most comprehensive event detection dataset of date, we acknowledge that there might be event types that we have overlooked. We plan to keep GLEN as a living project and update our ontology and dataset over time.

- Multilingual support. Currently, our documents are from the English PropBank annotation dataset so our system only supports English. One idea would be to utilize resources such as the Universal PropBank (Jindal et al., 2022)which can help us align corpora in other languages to PropBank and then further to our ontology.

## Acknowledgements

This research is based on work supported by U.S. DARPA KAIROS Program No. FA8750-19-2-1004. The views and conclusions contained herein are those of the authors and should not be interpreted as necessarily representing the official policies, either expressed or implied, of DARPA, or the U.S. Government. The U.S. Government is authorized to reproduce and distribute reprints for governmental purposes notwithstanding any copyright annotation therein.

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

## A Implementation Details

Table 6 presents the hyperparameters for three components in our model. During trigger identification, we restrict token spans to a maximum length of 10 tokens. For event type ranking, we use a designed loss function with $\tau = 1.0$. The top 10 ranked events are selected as candidates for event type classification. In event type classification, we do one round of self-labeling. Our model is trained on a single Tesla P100 GPU with 16GB DRAM.

| Component | TI | ETR | ETC |
|---|---|---|---|
| Training epochs | 5 | 5 | 2 |
| Batch size | 128 | 64 | 32 |
| Max sequence length | 128 | 128 | 512 |
| Base Model | Bert-base-uncased | | |
| learning rate | 1e-5 | | |
| Weight decay | 0.01 | | |
| Scheduler | Linear (with 50 warmup steps) | | |

Table 6: The training hyper-parameters of our model. **TI**: Trigger Identification. **ETR**: Event Type Ranking. **ETC**: Event Type Classification.

## B Baseline Implementation Details

We list the details of our baselines as below:

1. **Token Classification**: We use the IO tagging scheme to classify tokens.

2. **Span Classification**: We use the embedding of the first and last token to represent the span for classification.

3. **DMBERT** (Wang et al., 2019) is a BERT-based model that applies dynamic pooling (Chen et al., 2015) according to the candidate trigger's location. We consider single tokens to be candidate triggers in our dataset during testing.

4. **ZED** (Zhang et al., 2022) is an event detection model that utilizes definitions. Instead of pre-training on WordNet, we train the model on our noisy training data. As ZED only performs classification, we report the results with the trigger spans predicted by the token classification model (TokCls+ZED).

5. **InstructGPT**(Ouyang et al., 2022), also referred to as GPT-3.5, is the improved version of GPT3 trained with instruction-tuning. We use the text-davinci-003 model through the OpenAI API. We provide the model with an instruction for the task and 32 training examples from our training set for in-context learning.

We list the hyperparameters for our baseline models in Table 7. For ZED, we follow the original paper and set the margin to 0.2. We set the threshold for predicting an event type to 0.3 (if the cosine similarity between the event type representation and the trigger representation is smaller than this value, we will refrain from predicting any event type).

For the InstructGPT baseline, we use the text-davinci-003 model with a temperature of 0.2 and top_p set to 0.95 for decoding. We show our detailed prompt in Figure 8. The first part of the prompt is the task instruction and then we include 32 input-output examples. Due to the current input length limit of InstructGPT, we were unable to feed the ontology into the model as part of the input.

## C Data Filtering and Annotation Details

Table 8 shows some examples of removed Qnodes from DWD Overlay in our ontology with different kinds of reasons.

To improve dataset quality, we perform sentence de-duplication, remove sentences with less than 3 tokens and omit special tokens (marked by * or brackets). We ensure that every trigger is a continuous token span and we remove events with overlapping triggers. For the AMR dataset, we additionally remove triggers with a part-of-speech tag of MD (modal verbs) or TO (the word "to") (such cases do not appear in other datasets).

Based on this distant supervision dataset, we make further adjustments to the ontology. We manually inspected the most popular rolesets that have more than 1000 event mentions and removed rolesets that are too general or ambiguous (for instance, cause.01 and see.01). Finally, we remove the rolesets that have less than 3 event mentions across all datasets.

Figure 9 displays our annotation interface. The left box features the context, highlighting a single trigger, while the right box enumerates candidate event types, expressed as a combination of name and description, with an extra choice labeled "None of the above options is correct." Each Qnode is represented by its name and description. The number of options varies from 2 to 9 based on the ontology.

| Model | DMBERT | SpanCls | TokCls | ZED |
|---|---|---|---|---|
| Training epochs | 10 | 10 | 10 | 10 |
| Batch size | 64 | 64 | 16 | 16 |
| Negative samples | 5 | 5 | - | 5 |
| Learning rate | 5e-5 | 5e-5 | 2e-5 | 2e-5 |
| Max sequence length | | 64 | | |
| Base Model | | Bert-base-uncased | | |
| Weight decay | | 0.0 | | |
| Scheduler | | Linear | | |

Table 7: Hyperparameter settings for baseline models.

```
List the events that are mentioned in the given context and highlight the most related word or phrase
using the  tag for each event. There might be more than one event per context provided or no event
in the context.
For each tagged event, also generate its corresponding event type.
```

```
Context: The average yield on threemonth jumbos rose to 8.00 % from 7.96 % , while the two - year average
fell by the same amount to 7.89 % .

Output: The average yield on threemonth jumbos rose to 8.00 % from 7.96 % , while the two -
year average fell by the same amount to 7.89 % .

Events: rose:climb;fell:descent;

Context: i also think its stupid that our european friends can immigrate to britian and seek benefits , and
denying them that would be against " human rights " .... according to the eu court of human rights .

Output: i also think its stupid that our european friends can immigrate to britian and seek
benefits , and denying them that would be against " human rights " .... according
to the eu court of human rights .

Events:
immigrate:human_migration;denying:denial;against:anti-discrimination;

Context: Most of the 34 sailors hurt , are being reunited with family in Norfolk , Virginia , before moving
on to a naval hospital for further treatment .

Output: Most of the 34 sailors hurt , are being reunited with family in Norfolk , Virginia ,
before moving on to a naval hospital for further treatment .

Events:
hurt:damage;moving:active_motion;
```

Figure 8: Truncated version of our prompt to InstructGPT.

The annotator's task is to select the option that most accurately represents the trigger word.

Each instance was annotated by two annotators separately. For PropBank rolesets that were frequently labeled as "None of the above", we performed manual inspection to determine if the mapping should be removed or revised. Finally, for the affected instances and the instances with disagreement, we asked our in-house annotators to perform a third pass as adjudication.

| Qnode | Name | Roleset | Description |
|---|---|---|---|
| *Removed during cleaning the heavily used rolesets* | | | |
| Q2536390 | abdominal_distention | ill.01 | Physical symptom |
| Q192989 | acculturation | change.01 | process of cultural and psychological change |
| Q422268 | actinomycosis | ill.01 | Human disease |
| Q1319035 | adult_education | educate.01 | form of learning adults engage in beyond traditional schooling |
| Q9363879 | stamping | make.01 | metalworking |
| Q615857 | stapedectomy | surgery.01 | surgical procedure of the middle ear performed to improve hearing |
| Q366774 | adrenalectomy | remove.01 | surgical removal of the adrenal gland |
| Q2035485 | subcutaneous_injection | inject.01 | Medical procedure |
| *Removing reason: cognitive events that do not involve any physical state change* | | | |
| Q241625 | wish | wish.01 | desire for a specific item or event |
| Q26256512 | want | want.01 | economic term for something that is desired |
| Q26253999 | yearning | yearn.01 | deep and aching desire for someone or something |
| Q706622 | intention | intend.01 | mental state representing commitment to perform an action |
| Q3027692 | differentiation | differentiate.01 | process by which two closely related linguistic varieties diverge from one another during their evolution |
| Q104776298 | crosspatch | grouch.01 | a person who is easily annoyed |
| Q516519 | suspicion | suspect.01 | emotion |
| Q659974 | trust | trust.02 | assumption of and reliance on the honesty of another party |
| *Removing reason: mapped to too general or ambiguous rolesets* | | | |
| Q105606485 | intellectual_activity | think.01 | human activity comprising of mental actions |
| Q2944236 | photosensitivity | see.01 | Light sensitivity in homo sapiens |
| Q9174 | religion | believe.01 | set of beliefs, practices and traditions for a group or community |
| Q16513426 | decision | decide.01 | result of deliberation |
| Q2827815 | international_aid | give.01 | voluntary transfer of resources from one country to another |
| Q9081 | knowledge | know.01 | experience or education by perceiving, discovering, or learning |
| Q1221208 | employment_contract | agree.01 | agreement between employer and employee on terms of work and compensation |
| Q56274009 | looking | look.01 | act of intentionally focusing visual perception on someone or something |
| *Removing reason: low frequency* | | | |
| Q379788 | advection | advect.01 | transport of a substance by bulk motion |
| Q381105 | aeration | aerate.01 | process of circulating or mixing air with water |
| Q104541 | aerosol | aerosolize.01 | colloid of fine solid particles or liquid droplets, in air or another gas |
| Q623179 | state_terrorism | terrorism.03 | acts of terrorism against individuals conducted by organs of a state |
| Q98394474 | stenciling | stencil.01 | artistic technique for transferring images using stencils |
| Q844613 | sintering | sinter.01 | process of forming material by heat or pressure |
| Q249697 | eulogy | eulogize.01 | speech in praise of a person, usually recently deceased |
| Q901882 | interface | interface.01 | boundary between different phases of matter |

Table 8: Examples of removed Qnodes in XPO Overlay. Note that one node can be removed due to multiple reasons.

## D Impact of Self-labeling

| Roleset Category | Clean | Covered | Other | Total |
|---|---|---|---|---|
| #Instances | 3642 | 3308 | 1223 | 8173 |
| Hit@1 before | 95.74 | 48.52 | 37.37 | 67.89 |
| Hit@1 after | 95.47 | 54.59 | 39.17 | 70.50 |

Table 9: Hit@1 scores before and after self-labeling on different categories of PropBank rolesets. **Clean**: rolesets with only one candidate labels. **Covered**: rolesets covered in the self-labeled training data.

Figure 10 indicates that with a threshold[10] of 0.9, the accuracy of selecting the correct label from a candidate set reaches 57.8% on the dev set. To investigate how self-labeling contributes to the improvements, we categorize test instances into three groups based on their PropBank rolesets, as shown in Table 9. The 'Clean' rolesets map to only one event type in DWD Overlay. We train the base classifier on 71,834 training instances corresponding to these 'Clean' rolesets, which naturally performs significantly well on this portion of data. The model after self-labeling is trained with an additional 25,549 self-labeled data. The rolesets corresponding to these data are categorized as "Covered". Table 9 indicates that the main performance gain comes from the "Covered" data, which is boosted directly by including corresponding training data. The "Other" category also sees some improvement, at the cost of a slight drop in the "Clean" category.

---

[10]The threshold is the margin between the probability of the top 1 event type and the other types in a candidate set. A higher threshold means a higher level of confidence.

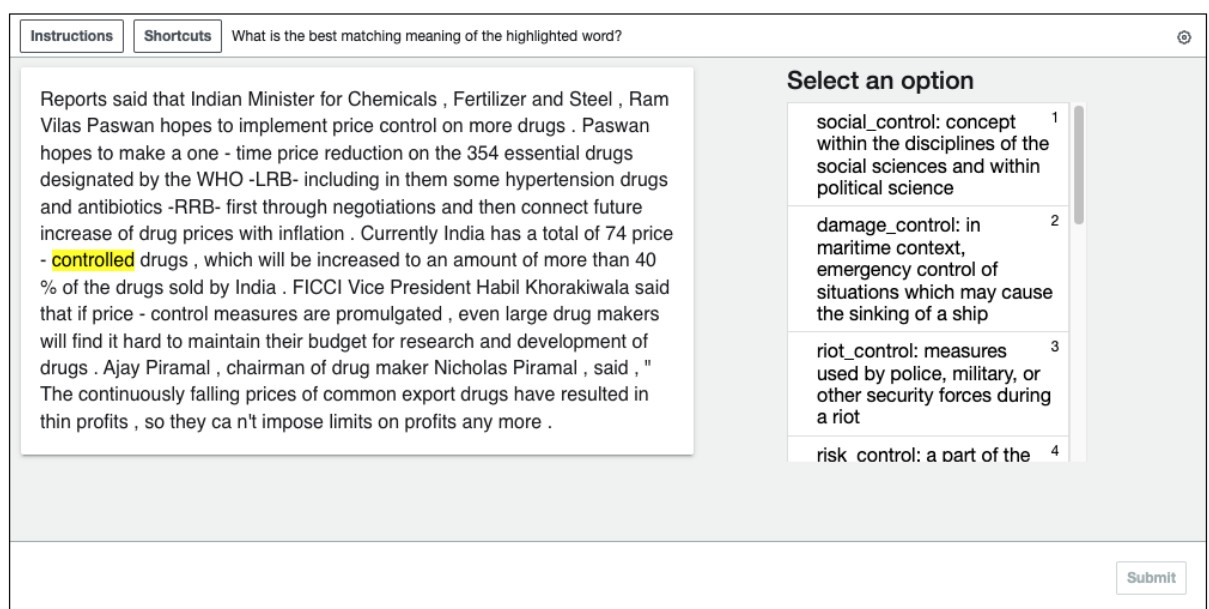

Figure 9: Annotation interface built with Amazon Mechanical Turk for labeling the development and test set.

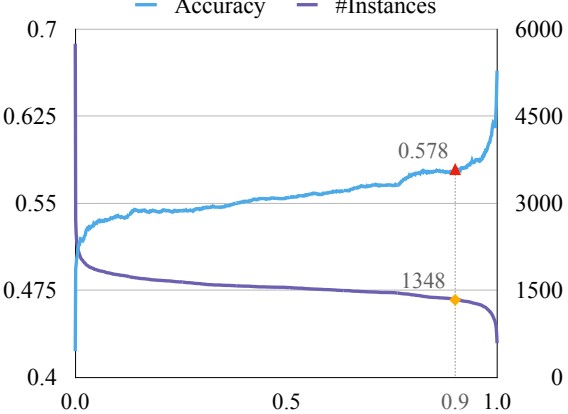

Figure 10: Relationship between the threshold and two metrics: the accuracy of label selection on dev set and the number of selected instances. The x-axis represents the threshold.