# OpenReview forum: "GLEN: General-Purpose Event Detection for Thousands of Types"
_EMNLP/2023/Conference — EMNLP 2023 Main_

### Official Review · Reviewer_ohVy · 2023-08-03

**Typos Grammar Style And Presentation Improvements:** Generally good grammar and presentati…
**Soundness:** 5

**Excitement:**

5: Transformative: This paper is likely to change its subfield or computational linguistics broadly. It should be considered for a best paper award. This paper changes the current understanding of some phenomenon, shows a widely held practice to be erroneous in someway, enables a promising direction of research for a (broad or narrow) topic, or creates an exciting new technique.

**Missing References:**

I could not detect missing references.

**Paper Topic And Main Contributions:**

The paper is well-written and well illustrated. Figure 2 clearly explains the process of turning the collected triggers into predicted results.

The main topic of the paper is making event extraction pipelines more accesible. A larger dataset with over 200k event mentions is created by using the DWD Overlay which provides mappings between Wikidata and PropBank. The results are generally better than InstructGPT and several older BERT models.

The paper starts by observing a disturbing fact: namely that the most used event extraction is almost two decades old. This is probably observed based on citations, which makes sense. However, given the advances in SoA, one would expect that annotation guidelines have changed a lot in the mean time. One would also expect that old datasets offer no surprises any more. This turns out to not be the case.

The paper is written in an unusual format, the Related Work section being a mere afterthought before the Conclusions section. This is extremely difficult to read by the casual reader. As a person familiar with NER/NEL/Slot Filling/Event Extraction I can jump directly into the topic and experimental section, but a casual reader that is not familiar with the topics may need some time to adjust. The classic structure of: Introduction, Related Work, Method, Experiments, Discussion (not only discussion around experimental results, but also around the previous work), Conclusion, Limitation is rather more suited for any paper. In fact I remember a Twitter thread from 2020-2021 that clearly explained this and was heavily retweeted.

Quantitative evaluation results showcase good results - though the evaluated model is simply mentioned as "Ours". This is confusing as future work will maybe call the model with a name that's not mentioned anywhere in the paper creating a source of confusion.

Table 3 could have used more rows. For example, it is not clear what happens when multiple triggers are in the same region. Will they all be collected and turned into good predictions?

The analytical section is rather good, but only provides examples for several error categories.

I have read the rebuttal and think they have addressed my concerns.

**Questions For The Authors:**

Q1: it is not clear what happens when multiple triggers are in the same region. Will they all be collected and turned into good predictions?

**Reasons To Accept:**

- creation of a huge dataset (GLEN) which is almost 2x the size of the previous largest event extraction dataset (MAVEN)
- development of good trigger identification and event type ranking methods
- strong analytical section
- the system will be released as a Docker container, thus enhancing reproducibility

**Reasons To Reject:**

- paper is written in an unusual format, leaving related work as one of the later sections of the paper and confusing casual readers
- section 5.3 about remaining errors is confusing - remaining after what?

**Reproducibility:**

5: Could easily reproduce the results.

**Reviewer Confidence:**

5: Positive that my evaluation is correct. I read the paper very carefully and I am very familiar with related work.

---

> ### Author Rebuttal · Authors · 2023-08-29
>
> Thank you for your detailed and insightful comments.
>
> While we did start the paper by mentioning that ACE05 is an old dataset, we don’t think being old itself is the biggest fault. The real concern is that due to the very restricted ontology of ACE, models that are trained on it cannot be used for real applications. This hinders the adoption of event extraction models, especially in comparison to other IE models, such as NER and entity linking.
>
> “Given the advances in SoA, one would expect that annotation guidelines have changed a lot in the meantime. One would also expect that old datasets offer no surprises any more” -- interestingly enough, the annotation guidelines have not changed since the majority of papers still use the original annotations in ACE, but newer work has found new settings such as few-shot learning, zero-shot learning or continual learning and adopted the old dataset for such cases. These efforts contribute to the long-reigning popularity of ACE and are orthogonal to our work.
>
> **Regarding model name and the analysis tables**: we think your suggestions are very useful and we will incorporate them in the revised paper. We actually selected examples for every category in the error analysis but had to comment some out to fit the paper limit. We will add them back given the extra page available.
>
> **Title of section 5.3**: we meant to say that these were “remaining errors” after our full model, as opposed to section 5.1 where we show error per stage. We will change this into “error categorization” or “error composition” to be more clear.
>
>
> **Regarding the “unusual paper structure”**: perhaps we were biased by pieces such as “How to Write a Great Research Paper” https://simon.peytonjones.org/great-research-paper/ which advocated for a structure that put related work towards the end of the paper. That being said, we appreciate your feedback on how delaying the Related Work section would cause difficulty to the causal reader. Besides moving the Related Work section after the introduction, we will also add some definitions for the event detection task to make the paper more approachable.
>
> Q1: Yes, our model can detect multiple triggers per sentence. In the trigger detection stage, we perform classification over all candidate spans that are less than 10 token long. In the event type ranking stage, although the ranking is done on the sentence-level, we take top-10 event type candidates, which can be mapped to different triggers in the sentence. Then finally, in the event type classification stage, we perform classification for each of the identified trigger spans in the sentence.
>
> For example, in the sentence “later she says intimidated into changing her vote to not guilty”, our model can detect 3 events with triggers “intimidated [Q3858303]”, “changing [Q1150070]” and “vote[Q189760]”.

---

### Official Review · Reviewer_rTom · 2023-08-06

**Soundness:** 4

**Excitement:**

4: Strong: This paper deepens the understanding of some phenomenon or lowers the barriers to an existing research direction.

**Paper Topic And Main Contributions:**

SUMMARY. The paper focuses on event detection/extraction (EE) - specifically, trigger detection and event type classification - in less investigated scenarios involving large, domain-general event type ontologies. For these scenarios, authors provide a dataset named "GeneraL-purpose EveNt benchmark (GLEN)" combining a large-scale Wikidata-based event ontology derived from the target of DWD Overlay mappings, with a large-scale corpus of event annotations derived by mapping PropBank gold standard event annotations to ontology types, obtained leveraging DWD Overlay and manual annotation for dev/test sets. Authors also propose a neural method (based on pre-trained LM - BERT) for trigger detection and type classification tailored to large event ontologies, and evaluate it on GLEN demonstrating superior performance w.r.t. applicable state-of-the-art methods.

CONTRIBUTIONS:

* C1. Domain-general, large-scale Wikidata-based event ontology grounded on PropBank, derived via manual cleaning from DWD Overlay mappings.

* C2. Domain-general, large-scale "GLEN" EE dataset derived from ProbBank gold standard annotations involving manual annotation (for dev/test sets).

* C3. EE method tailored to considered EE context.

* C4. Analysis of the GLEN dataset.

* C5. Extensive experiments investigating the performance of proposed method and relevant state-of-the-art systems on the proposed dataset

POST-REBUTTAL UPDATE. I thank the authors for the clarifications in their rebuttal. After reading it and considering the other reviews as well, I confirm the original scores in my review.

**Questions For The Authors:**

* A. How does the removal of cognitive Qnode events make the ontology "more suitable for the event extraction task" [§2.1]?

* B. What's the motivation for removing general / ambiguous PropBank rolesets like cause.01 and see.01 [§C, §2.1 "another round of data filtering"]? Is that there would be too many options for annotators to choose among?

* C. Does the error analysis for type classification [§5.3] refer exclusively to event triggers in ground truth data? If so, it would seem to me that error categories 'candidate set' (e.g., have trigger 'working' for roleset 'work.01', and predict wrong Qnode for 'work.01') and 'extended roleset' (e.g., for same example, predict a Qnode for 'work.XY' with XY != 01) should cover all possible error scenarios, as the trigger should constrain the possible Qnodes that can be picked for event classification (via 'trigger' -> possible rolesets for trigger -> possible Qnodes for such rolesets). Is it that Qnodes unrelated to trigger possible rolesets are considered as candidates following type ranking? Anyhow, it seems to me that error classes 'child', 'parent', 'sibling', and possibly 'unrelated', may form another orthogonal way to classify errors interesting fro practical applications (e.g., a user may still be satisfied if a child type is wrongly chosen since the parent type can be still inferred).

**Reasons To Accept:**

* S1. The GLEN dataset and ontology resources contributed by the paper, which involve substantial manual annotation work and distinguish themselves from existing EE resources for the large number of events covered, filling this way a gap and supporting further work in this context.

* S2. The proposed EE method which is shown to substantially improve over state-of-the-art methods in the considered EE context.

* S3. Informative experiments and analysis reported in the paper (including appendix).

* S4. Reproducibility facilitated by adequate description of the approach in the paper and by code and data provided as part of supplemental material.

* S5. Overall well written paper.

**Reasons To Reject:**

* W1. Event roles appear to be neglected in GLEN dataset and ontology, and arguments extraction and their assignment to roles is not part of the proposed extraction method.

* W2. Some choices related to the construction of the GLEN ontology are not completely clear (see questions A, B) and I wonder whether this may pose issues in using in practice GLEN (in its current version, as authors mention to evolve it) due to have discarded relevant types of events.

**Reproducibility:**

5: Could easily reproduce the results.

**Reviewer Confidence:**

3: Pretty sure, but there's a chance I missed something. Although I have a good feel for this area in general, I did not carefully check the paper's details, e.g., the math, experimental design, or novelty.

**Typos Grammar Style And Presentation Improvements:**

* T1.  [§2.1] "mainlly" -> "mainly"
* T2.  [§2.3] "wider range" -> "wider range than ..." or simply "wide range"

---

> ### Author Rebuttal · Authors · 2023-08-29
>
> Thank you for your detailed comments. We are glad that you find our dataset useful, our experiments informative and our paper well-written.
>
> We would like to provide our response to your mentioned weaknesses and questions as follows:
>
> W1: The GLEN dataset currently does not include argument roles and this is why we refer to it as a “General-Purpose Event Detection Dataset” instead of “Event Extraction Dataset” and we only state our contributions to event detection (Line 97-104). This is the same as MAVEN, which is meant to be a “A Massive General Domain Event Detection Dataset”. We do agree that argument extraction is an indispensable part of events and we are working on extending GLEN to cover arguments. We will make the distinction between event extraction and event detection more clear in our introduction and state the scope of our work more explicitly.
>
> QA: Following the convention of the most popular event extraction datasets ACE and ERE, we focus on events that are action-related and result in state changes in the physical world. Cognitive events only result in changes of the mental state and are more related to the opinion of the subject, which is related to other tasks such as stance detection.
>
> QB: This final round of filtering was done in a very restricted way, and on a case-by-case basis through discussion. We took the list of the most frequent rolesets and went over the examples to see if these rolesets were suitable to serve as event types. `see.01` was overloaded and it was used for both the physical sense of see ( I saw a tree out the window) and the metaphorical sense of see ( I see a bright future for him). Since the latter does not involve any physical state changes, we deemed that see.01 was not suitable to keep as an event type. For `cause.01`, this is because causal relations are usually a separate topic of research (such as in [1]) and usually not included as an event type.
>
> QC: Yes, Qnodes unrelated to trigger possible rolesets are still considered as candidates following type ranking, we did not use the ontology to do another round of filtering. In addition, some of the ground truth types did not appear in the top-10 ranked types (this accounts for ~10% of the errors as shown in Table 4).
> The mapping between Qnodes and rolesets is many-to-many, so the candidate rolesets cannot be directly determined by taking the lemma of the trigger. For example “form.01”, “create.01”, “make.01” and “craft.01” are all mapped to the Qnode “creation”.
> We agree that the two ways of categorizing errors are a bit orthogonal (one is based on the mapping, another is based on the ontology hierarchy) and we can split them into two pie graphs for analysis.
>
>
> [1] Modeling Document-level Causal Structures for Event Causal Relation Identification (Gao et al., NAACL 2019)

---

### Official Review · Reviewer_7vXC · 2023-08-12

**Soundness:** 4

**Excitement:**

4: Strong: This paper deepens the understanding of some phenomenon or lowers the barriers to an existing research direction.

**Paper Topic And Main Contributions:**

This paper build a new event detection dataset named as GLEN using distant supervision and based on the DWD Overlay method to automatically map Wikidata to PropBank rolesets so that the event triggers in sentences can be automatically annotated. Apart from building the new dataset, the authors also proposed a multi-stage event detection method.

The new dataset address the problem that there lack large-scale, diverse event detection dataset, the proposed approach is designed to address that difficulties met when working on this new dataset.

**Questions For The Authors:**

-  Is the manually cleaned up of events fime consuming? (line 128)
- Do you have analysis on error propagation? The multi-stage approach compared with end-to-end approach would suffer from error propagation more
- Have you tried with concatenate the sentence and event description? The current approach is using different embeddings for these two and then calculate similiarty, it might cause alignment issue

**Reasons To Accept:**

- contribute directly to the event detection system by releasing a new large scale dataset with 3465 fine-grained event types and 205k event instances
- proposed new method of event detection
- although annotation method is an already existing method, "DWD Overlay", the authors have their filtering of the event types and efforts on collecting and cleaning the data
- ontology is a superset of ERE and ACE, which is an advantage for fair comparison

**Reasons To Reject:**

- the annotation method is a previous work "DWD Overlay", but the authors have explained the difference
- the quality of dataset is not explicit, if possible, a histogram of distributtion of data types will be more helpful than Figure 3

**Reproducibility:**

4: Could mostly reproduce the results, but there may be some variation because of sample variance or minor variations in their interpretation of the protocol or method.

**Reviewer Confidence:**

4: Quite sure. I tried to check the important points carefully. It's unlikely, though conceivable, that I missed something that should affect my ratings.

**Typos Grammar Style And Presentation Improvements:**

- line 481, type "toward" should be "towards"

---

> ### Author Rebuttal · Authors · 2023-08-29
>
> Thank you for appreciating our effort in creating GLEN: including the ontology, the dataset, and the corresponding system.
>
> We would like to clarify the differences between GLEN and the existing DWD ontology. The DWD ontology is a human-curated subset of WikiData nodes, each entry in this ontology includes properties from WikiData (name, definition, overlay_parent and similar_nodes), properties from PropBank (argument names) and additional mappings to LDC event types (ldc_types). The DWD ontology can be seen as a dictionary,  but we believe it would be quite a stretch to say that the dictionary itself is an “annotation method”. We will add more description about the DWD ontology in Section 2.1 to make this more clear.
>
> We used Figure 3 mainly to display the long-tail property of the dataset, showing the relationship between the frequency of the event type and the rank of the event type, as in Zipf’s law.
> A histogram would also be good for showing the dataset distribution and while we cannot attach figures here, we can add one in our Appendix during revision.
>
> Regarding the dataset split, since our dataset is much larger than ACE or ERE, having a higher proportion of test/dev instances would require much more human annotation. Although we simplified the annotation task by reusing the trigger spans, our experience with MTurk was still quite bad initially and we had to filter many annotators, redo part of the annotation, and then ask in-house annotators to adjudicate when there was disagreement. Such issues made it quite difficult to scale up the test/dev instances.
>
> To address your questions:
>
> **For the cleanup procedure mentioned in Line 128**:
> in terms of heavily reused rolesets, we manually inspected 47 rolesets from Propbank corresponding to more than 10 DWD Overlay nodes. These nodes were either discarded or relabeled. One of our authors did an original pass of the labeling and then we asked some of our in-house annotators to check. A single pass took a few hours to complete.
>
> **Analysis of error propagation.**
> We provided per-stage results in Table 2 and Table 4. The three stages of our method lead to three possible sources of errors: (1) trigger detection errors, which is shown in Table 2 Trigger Identification stage; (2) type ranking errors, as shown in Table 4 row 1; (3) type classification errors, as shown in Table 4 row 2. Among these three types of errors, trigger identification is the most prominent source. Although our pipelined model does have some error propagation issues, this does not put us at a disadvantage compared with the single stage classification baselines as they show even lower trigger identification scores.  (ZED is also a pipelined method.)
>
> **Concatenation of the sentence and the event description.**
>  As mentioned in Section 3.3 Line 282-285, the third stage of our model, the type classification stage, takes the concatenated event description and sentence as input. “Using separate embeddings for these the event and the sentence and then calculate similarity” is only for our event ranking stage (Line 248-266). We agree with your intuition that separate encoding usually does not lead to the best performance, but it is more efficient, which is exactly the motivation behind our multi-stage approach (Line 143-147). We did not directly compare against "skipping type ranking and only using type classification" because it would require encoding each sentence 3500 times (for each trigger) to perform inference, which would be prohibitively slow.

---

### Meta-Review · Area_Chair_3DGV · 2023-09-21

**Recommendation:** 5

**Metareview:**

The GeneraL-purpose EveNt benchmark (GLEN) is a notable initiative aimed at advancing event benchmarking. It achieves this by combining a substantial Wikidata-based event ontology, derived from DWD Overlay mappings, with a comprehensive corpus of event annotations. These annotations are created through the mapping of PropBank gold standard event annotations to ontology types, augmented by manual annotation for dev/test sets. As a result, GLEN introduces a novel dataset and presents an innovative event detection method, establishing the first baseline for this dataset. The associated research paper is well-written and effectively illustrated, but it would benefit from reorganizing the related work section to provide better context.

Strengths:

* Innovative Approach and Large-scale Corpus: GLEN introduces a novel dataset and event detection method, contributing to the field of event benchmarking.
* Well-Written Paper: Oveall the paper is well written and easy to follow.

Weaknesses:

* Quality Assessment: GLEN lacks explicit quality assessment metrics for its dataset, such as a histogram showcasing the distribution of data types.
* Limited Scope: The focus of GLEN is primarily on event detection, neglecting event roles, which might limit its applicability in certain contexts.
* Typos and Structure of the Paper: The paper contains a few typographical errors that could affect the overall readability and keeps related work at the end making it hard for casual readers to follow the work.

---

### Decision · Program_Chairs · 2023-10-07

**Decision:**

Accept-Main

**Comment:**

The GeneraL-purpose EveNt benchmark (GLEN) is a notable initiative aimed at advancing event benchmarking. It achieves this by combining a substantial Wikidata-based event ontology, derived from DWD Overlay mappings, with a comprehensive corpus of event annotations. These annotations are created through the mapping of PropBank gold standard event annotations to ontology types, augmented by manual annotation for dev/test sets. As a result, GLEN introduces a novel dataset and presents an innovative event detection method, establishing the first baseline for this dataset. The associated research paper is well-written and effectively illustrated, but it would benefit from reorganizing the related work section to provide better context.

Strengths:

* Innovative Approach and Large-scale Corpus: GLEN introduces a novel dataset and event detection method, contributing to the field of event benchmarking.
* Well-Written Paper: Oveall the paper is well written and easy to follow.

Weaknesses:

* Quality Assessment: GLEN lacks explicit quality assessment metrics for its dataset, such as a histogram showcasing the distribution of data types.
* Limited Scope: The focus of GLEN is primarily on event detection, neglecting event roles, which might limit its applicability in certain contexts.
* Typos and Structure of the Paper: The paper contains a few typographical errors that could affect the overall readability and keeps related work at the end making it hard for casual readers to follow the work.